# Physical Exercise and Selective Autophagy: Benefit and Risk on Cardiovascular Health

**DOI:** 10.3390/cells8111436

**Published:** 2019-11-14

**Authors:** Ne N. Wu, Haili Tian, Peijie Chen, Dan Wang, Jun Ren, Yingmei Zhang

**Affiliations:** 1Department of Cardiology, Zhongshan Hospital, Fudan University, Shanghai 200032, China; 14301050124@fudan.edu.cn; 2Shanghai Institute of Cardiovascular Diseases, Department of Cardiology, Zhongshan Hospital, Fudan University, Shanghai 200032, China; 3School of Kinesiology, Shanghai University of Sport, Shanghai 200438, China; tianhaili123@163.com (H.T.); chenpeijie@sus.edu.cn (P.C.); 4School of Physical Education and Sport Training, Shanghai University of Sport, Shanghai 200438, China; danwang_dana@163.com

**Keywords:** cardiovascular, physical exercise, autophagy, selective autophagy, benefit, risk

## Abstract

Physical exercise promotes cardiorespiratory fitness, and is considered the mainstream of non-pharmacological therapies along with lifestyle modification for various chronic diseases, in particular cardiovascular diseases. Physical exercise may positively affect various cardiovascular risk factors including body weight, blood pressure, insulin sensitivity, lipid and glucose metabolism, heart function, endothelial function, and body fat composition. With the ever-rising prevalence of obesity and other types of metabolic diseases, as well as sedentary lifestyle, regular exercise of moderate intensity has been indicated to benefit cardiovascular health and reduce overall disease mortality. Exercise offers a wide cadre of favorable responses in the cardiovascular system such as improved dynamics of the cardiovascular system, reduced prevalence of coronary heart diseases and cardiomyopathies, enhanced cardiac reserve capacity, and autonomic regulation. Ample clinical and experimental evidence has indicated an emerging role for autophagy, a conservative catabolism process to degrade and recycle cellular organelles and nutrients, in exercise training-offered cardiovascular benefits. Regular physical exercise as a unique form of physiological stress is capable of triggering adaptation while autophagy in particular selective autophagy seems to be permissive to such cardiovascular adaptation. Here in this mini-review, we will summarize the role for autophagy in particular mitochondrial selective autophagy namely mitophagy in the benefit versus risk of physical exercise on cardiovascular function.

## 1. Introduction

Regular physical exercise is a part of healthy lifestyle, with multiple cross-sectional studies consolidating reduced overall risk of cardiovascular diseases and cardiac events associated with habitual or leisure physical exercises [1,2]. Ample evidence has indicated a much better survival rate following a cardiovascular event in those who are physically active in comparison with more sedentary individuals, and the beneficial impact of physical exercise on heart failure is also described [1,3,4,5]. Regular physical exercise is now becoming a non-pharmacological remedy to lower cardiovascular morbidity and mortality courtesy of the exercise-induced cardiovascular benefit [6,7]. Such maneuver drastically improves the overall cardiovascular survival despite the poor success for current pharmaceutical therapeutics against cardiovascular diseases. More evidence has favored a close dose–response correlation between exercise duration and/or intensity and overall cardiovascular benefit [8,9,10]. Although the precise nature of physical activity in cardiovascular regulation and disease prevention remains poorly defined, low-, moderate-, and vigorous-intensity exercise have all exhibited some degrees of health benefit [1,11,12]. Moreover, it has been reported that intense/extreme exercise may also be detrimental to human hearts [1,8]. Here the term ‘exercise’ is mainly used to reflect regular aerobic or endurance exercise, unless otherwise stated.

To-date, a number of theories at the cellular and molecular levels have been postulated towards exercise-offered cardiovascular benefits including increased insulin sensitivity, reduced oxidative stress and adiposity, fiber transformation toward oxidative myofibers, and increased mitochondrial content/function [1,12,13]. More recently, there is a growing concern of physiological adaptation and induction of autophagy, a conserved evolutionary process responsible for the degradation of multiple cellular components [1,14,15,16]. Under stress conditions such as starvation and increased physical activity, autophagy is usually turned on to recycle long-lived or damaged cellular organelles and proteins for the resynthesis of new organelles and ATP. Particular recognition and degradation of damaged or superfluous organelles may also be achieved by a special form of autophagy—selective autophagy [17,18]. Recently, a growing body of literature has recognized the pivotal role of mitophagy—selective autophagy of mitochondria, in conditions with energy stresses, such as starvation, obesity and physical exercise [19,20]. In the heart, timely removal of dysregulated (long-lived or damaged) mitochondria is essential to cardiac homeostasis, while excessive or pathological mitophagy is deemed harmful to the organism [21,22,23,24,25,26]. Interestingly, previous studies indicated that not only does exercise transiently induce cardioprotective mitophagy, but also helps to sustain a proper level of mitophagy over time [15,27]. Thus, mitophagy might be instrumental to a better understanding of how exercise impacts the overall organismal health. In this mini-review, we will highlight the essential role of mitophagy in exercise-induced effects on cardiovascular system.

## 2. Contemporary Theory behind Exercise-Induced Cardiovascular Benefit

Physical exercise of sufficient intensity and duration improves cardiovascular performance and cardiac reserve in healthy individuals [28]. For example, a 3–5 day short-term endurance training may elicit cardioprotection against ischemia–reperfusion insult, although molecular mechanisms behind physical exercise-induced cardioprotection remain elusive. In-depth analysis was carried out by Luan and associates in an effort to recapitulate beneficial effects of various types of exercise on 26 forms of chronic diseases, including cardiovascular diseases. These authors have concluded that long-term aerobic or home-based exercise seems to benefit patients with coronary artery diseases the most, while high-intensity interval training (HIIT) significantly enhances cardiac performance in patients with chronic heart failure [12]. These notions are supported by more studies focusing on exercise, cardiovascular function, and structure, which will be discussed in detail below.

### 2.1. Exercise-Induced Functional and Structural Changes

At the tissue level, exercise-induced cardiovascular benefits can be divided into two broad types: Functional responses to higher energy demand during exercise and chronic adaptations in the long run. The latter is thought to be a predominant feature of sustained exercise (i.e., structured, well-planned, and repetitive physical activity). This section will begin with cardiac changes induced by short-term exercise. Then we will discuss chronic responses of physical exercise, mainly focusing on three themes—metabolic flexibility, cardiac remodeling, and angiogenesis (shown in Figure 1).

### 2.2. Acute Alterations in Cardiac Function during Exercise

With exercise, hearts will experience physiological adaptations including increased cardiac output (CO) and peripheral perfusion to cope up with dramatically increased musculoskeletal and pulmonary requirements [29,30,31]. The higher CO results from a concerted effort from increased heart rate (HR), stroke volume (SV), and/or cardiac contractile capacity [29,32]. In addition, exercise may stimulate autonomic function to promote cardiac function. Cardiac chronotropic and inotropic responses to sympathetic system (β-adrenergic response) may be facilitated by exercise along with stimulation of intrinsic myogenic tone [13]. Ample evidence has depicted a rather minor role for parasympathetic system in the tonic control of myocardial function, with norepinephrine from sympathetic nerve fibers being the predominant myocardial regulator in response to exercise [33]. Norepinephrine binds with β1 receptor to turn on G protein and adenylate cyclase. In consequence, cAMP is accumulated in the cytosolic space leading to elevated intracellular Ca^2+^ levels and higher cardiac contractility [34], which may also be arrhythmogenic and harmful if it is excessive.

### 2.3. Metabolic Flexibility

Metabolic flexibility refers to the ability of an organism to adapt changes in metabolic demand [35]. Physical exercise significantly increases energy expenditure and demand. Previous findings have identified a link between exercise and improved fatty acid and/or glucose oxidation [36,37,38]. During exercise, changes in mechanical stretch, catecholamines and circulating substrates (such as free fatty acids) impact cardiac metabolism. Glucose catabolism is transiently suppressed during exercise and is then elevated above the un-trained state after recovery [39]. In this regard, these metabolic changes are not only transient responses to physical activity but also adaptations that prepare the organism for the next bout of activity [40]. This is possibly achieved through autophagy and other cellular catabolic processes to elevate metabolism capacity [40]. Exercise also appears to improve insulin signaling. Exercise is known to promote insulin sensitivity and benefit glucose and energy homeostasis given that insulin signaling is vital for GLUT-4 and hemodynamic function [41]. Preserved glucose uptake has been documented in insulin-resistant muscle following exercise [42]. These events would promote glucose utilization and energy production in the heart. Mounting evidence has suggested that exercise may improve cardiovascular function through indirect actions on lipid and insulin profiles [11,43]. In addition, non-target GC-MS metabolomics analysis of rat hearts revealed that endurance training offered cardioprotection against ischemia-reperfusion injury possibly through modulating protein quality control, CoA biosynthesis and ammonia recycling [44]. Taken together, greater emphasis should be geared towards metabolic adaptations and mechanisms underlying metabolic flexibility, such as autophagy, during and after exercise.

### 2.4. Chronic Adaptations in Heart and Vasculature

Cardiac hypertrophy is thought to be a part of the adaptive remodeling process [5,45,46]. The heart mass, especially those within the ventricular wall (eccentric hypertrophy), rises physiologically as a result of sustained changes in metabolic and remodeling pathways in the heart [47]. Unlike hypertrophy observed in pathological conditions, such as hypertension, this cardiac hypertrophy is characterized by a mild increase in ventricular volume accompanied with reserved or increased myocardial function due to cardiomyocyte growth in size. In addition, this physiological hypertrophy displays none of the features of adverse cardiac remodeling, such as cardiac fibrosis and necrosis [48]. More recent studies have demonstrated distinct signaling molecules mediating cardiac hypertrophy in both physiological and pathological states [49], while how exercise exerts disparate induction of hypertension remains unclear. Induction of IGF-1/IRS-PI3K-Akt pathway is deemed to mediate physiological hypertrophy, which regulates several transcriptional factors [1].

In addition, intermittent hemodynamic stimuli induced by exercise also enhances vascular structure (i.e., increased angiogenesis) and function, which contribute to the increased cardiac output (CO) and lessened atherosclerosis [4]. Exercise training is probably the most sufficient way to improve endothelial function. Based on a systematic review and meta-analysis, a reduction in blood pressure was noted in patients of stroke or transient ischemic attack following exercise training [50]. Complex factors, such as shear stress and alternations in plasma profiles precipitate the activation or restoration of endothelial pathways during exercise [51]. Exercise-induced circulating catecholamines could act on β-3 adrenergic receptors (B3AR) to increase endothelial nitric oxide synthase (eNOS), which augments the bioavailability of NO (nitric oxide), an essential molecule responsible for vasodilation and anti-atherosclerosis effects [52,53,54,55]. More recent evidence suggested that rhythmic handgrip exercise promoted increased eNOS phosphorylation, NO generation, and O_2_^−^ production, along with improved autophagy markers including Beclin1, microtubule-associated proteins 1A/1B light chain 3B (LC3B), autophagy-related gene 3 (Atg3), and lysosomal-associated membrane protein 2A (LAMP2) as well as decreased levels of p62 in endothelial cells from human radial artery [56]. These findings denote a close tie between eNOS/NO signal cascade and autophagy in exercise-induced regulation on endothelial function.

### 2.5. Cellular and Molecular Alternations Induced by Exercise

At the cellular level, findings have indicated that physiological hypertrophy is accompanied with the induction of several mechanisms that promote cellular survive, including protein quality control, cell growth protein synthesis, antioxidant generation, autophagy-lysosomal system, and mitochondrial adaptation [1]. In a recent randomized controlled trial, endurance training and interval training (but not resistance training) were found to promote telomerase activity and telomere length, essential markers for cellular senescence, regenerative capacity, and healthy aging [57]. Moreover, physical exercise appears to exert a favorable effect on aging-related cardiometabolic stress through mediating autophagy [58].

Among these mechanisms mentioned above, emerging findings have consolidated a critical role for mitochondria in exercise-offered cardiovascular benefit. Mitochondrial remodeling is a vital determinant in exercise-dependent adaptations. Metabolic changes induced by exercise may influence mitochondrial function, dynamics and turnover, leading to robust mitochondrial network and enhanced metabolic flexibility. It has been shown that the transcription factor EB (TFEB) translocated to myonuclei during exercise and regulated mitochondrial biogenesis and glucose uptake, therefore acting as a major mediator for metabolic flexibility [59]. During exercise, there is a significant increase of mitochondrial biogenesis. Catabolic process through mitophagy is required to confer materials for synthesis and remove dysfunctional organelles that otherwise might result in cellular death. Thus, it is probable that the cardioprotective effects of exercise are strongly associated with mitophagy.

To further discern the upstream pathways in exercise-induced mitochondrial biogenesis and mitophagy, a number of studies were performed which have greatly enriched our knowledge of the impact of exercise on mitochondrial integrity [15,27,60,61,62,63]. For example, it was demonstrated that exercise-induced phosphorylation of an important energy sensor protein kinase AMPK (protein kinase AMP-activated catalytic subunit alpha 1) and AMPK-dependent ULK1 (unc-51 like autophagy activating kinase 1) phosphorylation is required to target lysosome to mitochondria [64]. Previous studies have recognized a rather pivotal role for transcriptional coactivator peroxisome proliferator-activated receptor-γ coactivator-1α (PGC-1α) in mediating exercise-induced responses on mitochondria. PGC-1α is capable of interacting with several nuclear transcription factors, such as peroxisome-proliferator activated receptor β (PPARβ) and estrogen-related receptor (ERR) to increase mitochondrial biogenesis and to improve mitochondrial energy metabolism [65,66]. Exercise restores mitophagy in high-fat high-fructose-treated liver in a PCG-1α-dependent manner [67], while deletion of PCG-1α compromises the flourishing of mitochondria following exercise [68]. However, Kang and Ji established an overexpression model of PCG-1α via in vivo transfection and found that PCG-1α overexpression drastically suppressed the levels of FoxO1/3 and mitophagy in immobilization-remobilization muscles [69]. Furthermore, the IGF-1/PI3K/Akt cascade implicates in chronic cardiac adaptations following exercise through regulating diverse cellular functions, such as cell growth, glucose metabolism and mitochondrial turnover [1,48]. Akt inhibits the transcription factor C/EBPβand then frees certain serum response factors (SRF) to bind target gene promotor, which orchestrates the maintenance of healthy mitochondrial network and contributes to cardiac hypertrophy [1,48]. Collectively, these studies have delineated general mechanism underlying exercise-induced mitophagy (shown in Figure 2), while more questions remain to be answered.

## 3. Risk of Exercise for Cardiovascular Function

Regular exercise provides benefit to cardiovascular function [70], while much uncertainty still exists with regards to the impact of strenuous exercise. To date, most studies assumed that whether exercise is salutary largely depends on the frequency, intensity, and duration of exercise [71]. High levels of physical exercise well beyond the recommended levels are tied with higher mortality risks in patients with preexisting cardiovascular diseases. Nevertheless, how much exercise is optimal to exert cardiovascular benefit remains unclear and equally controversial [72,73]. Recent studies have suggested a U-or J-shaped curve which reflects the association between exercise level and health outcomes [74,75]. Substantial evidence has shown that moderate levels of exercise are associated with a reduction in cardiovascular risks [47,76,77]. While too much exercise may be detrimental and is associated with increased risk of cardiovascular mortality [47,75]. Reports in endurance runners demonstrated that marathoners who completed at least 25 marathons in more than 25 years normally possess more severe coronary artery calcification and calcified coronary plaque [78]. A recent survey denoted that individuals who maintain a very high level of physical activity have likely higher odds of developing coronary artery calcification, especially in white American males [79]. Similarly, a large prospective cohort finding from Armstrong and colleagues involving 1,000,000+ women suggesting that strenuous daily physical activity may impose much higher risks of coronary heart disease [74]. Not surprisingly, we should take special precaution in weighing the overall benefit versus risk when advising individuals with regards to the physical exercise engagement. A number of unfavorable cardiovascular events may occur following intensive or excessive physical exercise. For example, exercise is known to precipitate angina pectoris, myocardial infarction, arrhythmias, and sudden death in those individuals with pre-existing coronary artery diseases [1,8,80].

There are emerging data denoting that sustained intense exercise may lead to adverse electrical and structural remodeling in the heart [81]. Moreover, plasma catecholamine responsiveness may be inappropriately affected by exercise which is manifested as chronotropic incompetence and lower plasma epinephrine response to exercise probably as a result of abnormal sympathoadrenal and autonomic function. Sustained exposure of catecholamine may trigger downregulation of β-adrenergic receptors (desensitization), resulting in loss of adenylate cyclase responsiveness and cardiac contraction during exercise. The β-adrenergic receptor-adenylate cyclase signaling cascade is essential to the maintenance of myocardial homeostasis [82]. A loss in either quantity or sensitivity of β-adrenergic receptors should disengage myocardium to sympathetic innervation (through norepinephrine) during physical exercise. Likewise, modification of β-adrenergic receptor-linked adenylate cyclase may also decrease adenylate cyclase activity and exercise capacitance. Therefore, decreased (or sometimes unchanged) myocardial contractile function during exercise fails to cope with the need from cardiopulmonary system for blood and oxygen for a homeostatic condition. Other than decreased left ventricular contraction, compromised diastolic function was also noted during exercise [83]. Although a number of mechanisms have been put forward, loss of myocardial function at rest and during exercise seems to be associated with myocardial alterations including myosin isozyme switch (V1 to V3) and phosphorylation of cardiac inhibitory protein TnI [34].

In contrast, this scenario may not hold true in healthy individuals. High levels of strenuous or vigorous exercise seem to have little effects on overall mortality in healthy individuals although intensive training may compromise the health benefits associated with regular moderate physical activity [72]. Greater emphasis should be made on how a well-functioning organism or individual combats the risks of exercise. Mounting efforts have illustrated exercise, especially intense or prolonged exercise, may cause oxidative stress and subsequent damage in myocytes [80]. Oxidative stress, energy requirement, and mitochondria are closely linked [20,84]. Therefore, we may propose that mitochondrial quality control is indispensable for beneficial adaptations induced by exercise. Oxidative stress could activate mitophagy to cope with mitochondrial dysfunction. Earlier studies have demonstrated a strong association between protective mitophagy and exercise, which we will elaborate in the next section.

## 4. Mitophagy and Exercise

Mitophagy is initiated when damaged mitochondria are labeled for degradation [20]. The major fission protein Drp1 (dynamin related protein 1) is translocated to depolarized mitochondrial membrane and segregates the damaged components from the rest of the healthy mitochondria [20,85]. Then, PINK1 (PTEN induced kinase 1) accumulates on compromised mitochondria and recruits E3 ubiquitin-protein ligase Parkin, which ubiquitinates a branch of proteins on outer mitochondrial membrane (OMM) [20,86]. Certain autophagy receptors, such as NDP52 (CALCOCO2, Ca^2+^ binding and coiled-coil domain 2) and optineurin then tether mitochondria to autophagosomes, which subsequently fuse with lysosomes for lysosomal degradation. It is noted that PINK1 would recruit autophagy receptors at a low rate independent of Parkin [87]. In addition to the PINK1/Parkin signaling cascade, several OMM-localized mediators, including NIX (NIP3-like protein X), BNIP3 (BCL2 interacting protein 3), FUNDC1, and cardiolipin could target mitochondria to autophagosome through binding to LC3 (microtubule associated protein 1 light chain 3α) on phagophores in response to developmental signals or hypoxia [88,89]. However, it should be noted that chronic hypoxia may overtly upregulate the level of housekeeper proteins. Thus, data normalized against these housekeeping proteins, such as GAPDH, actin and tubulin should be handled with special caution when heart tissue is exposed to hypoxia [90].

Exercise-induced mitophagy might slightly differ from the conventional pathways. It has been demonstrated that Parkin is indispensable for exercise-induced mitophagy initiation. Exercise stimulates mitophagy flux courtesy of increased recruitment of Parkin to mitochondria, despite that Parkin knockout did not impact basal mitophagy [91]. Examination conducted by Drake and colleagues found enhanced mitophagy levels in the absence of discernable PINK1 accumulation in skeletal muscles following exercise, while HeLa cells treated with carbonyl cyanide m-chlorophenyl hydrazone (CCCP) displayed overtly elevated PINK1 [92]. The relationship between exercise and mitophagy has been extensively studied, mainly using skeletal muscle or myocytes. Given the critical role of mitochondria in cardiomyocyte energy production and function [20], there has been an increasing interest in exercise-induced mitophagy in heart. In this section, we will introduce recent studies (last 5 years) on how exercise regulates mitophagy.

### 4.1. Exercise as a Treatment or Prevention to Diseases: The Role of Mitophagy

First, a mainstream of research has focused on revealing the close tie between exercise and temporarily enhanced mitophagy. It was indicated that Beclin1, LC3, and BNIP3 were remarkably upregulated in rat myocardium during acute exercise and were then slowly declined to baseline 48 h later [93]. Likewise, PINK1, Parkin, Ubiquitin, p62, and LC3 were overtly elevated in rat skeletal muscles after downhill treadmill running for 90 min with the upregulation lasting for more than 24 h [94]. It is noteworthy that shear stress has emerged as a modulator of autophagy during exercise. It was reported that 1 h of rhythmic handgrip exercise initiated autophagy, NO generation and O_2_^−^ production in humans due to the elevated shear stress [56]. In an earlier study, it was determined that inhibition of autophagy prevented NO production and enhanced ROS formation [95]. Thus, autophagy plays a critical role in NO bioavailability and redox homeostasis in endothelial cells. In addition to acute exercise, Ju and coworkers observed remarkable activation of autophagy flux and mitochondrial dynamics (both fusion and fission) in mice following sustained (8-week) swimming training. Moreover, when treated with colchicine, a blocker for autophagosomal degradation, BNIP3 was found increased while exercise-induced mitochondrial biogenesis was greatly diminished, indicating a possible role of mitophagy in mitochondrial content or biogenesis following exercise [96].

To date, studies have recognized a protective role of mitophagy during exercise. Mitophagy flux presumably protects the heart from exercise-induced risk. It is possible that mitophagy was stimulated by exercise-related activation of inflammation and accumulation of ROS, while upregulated mitophagy could remove ROS and eliminate inflammation, thus reducing mitochondrial injuries [93]. Figure 1 shows the possible schematic of how exercise exerts cardioprotective effects through modulating mitochondria homeostasis. Moreover, exercise shows promise as a safe and inexpensive way to treat multiple diseases, including cardiovascular diseases. There is an increasing emphasis on mitophagy in exercise treatment. Short-duration exercise regimen has been recommended for cardiac rehabilitation after stable myocardial infarction based on the favorable response of short-duration exercise (15-min swimming training per day, 5 times per week for 8 weeks) on cardiac function in mice. It has been suggested that increased SIRT3 as well as PINK1/Parkin was responsible for this [62]. Moreover, it was suggested that long-term (8 weeks) exercise coupled with caloric restriction prior to isoproterenol injection may prevent heart failure more efficiently than either therapy alone possibly through stimulation of autophagy [97]. Despite few data available on the role of mitophagy in resistance exercise, it was indicated that resistance exercise may attenuate muscle atrophy through elevated mitophagy and biogenesis in rats [98].

It is speculated that autophagy is required during caloric restriction and physical exertion for survival, and is repressed in nutrient-rich conditions [99]. However, human beings are no longer forced to be engaged in frequent physical activity in modern life, with the development of science and technology. Moreover, there is a rising concern that both sedentary behavior and caloric abundance are major contributors to a range of chronic diseases, including insulin resistance, obesity, diabetes mellitus, cardiovascular diseases, and various forms of cancer [100,101,102,103,104], while regular physical exercise helps to prevent these chronic diseases [105,106]. Moreover, metabolic diseases are among the major independent risk factors of cardiovascular diseases. Hopefully, physical exercise would promote cardiovascular health through primary and secondary prevention. Therefore, a number of investigators have sought to determine the salutary effects of exercise concurrent with low-quality diet. In particular, the contribution of autophagy or mitophagy has drawn close attention recently. Markers of mitophagy, autophagy, and mitochondrial dynamics were assessed in high-fat diet treated mice which were also engaged in either voluntary physical activity (VPA) or endurance training (ET). Researchers found that both VPA and ET rescued the high-fat-related increase of apoptosis and decrease of autophagy and mitochondrial biogenesis in mouse livers, leading to protection against nonalcoholic steatohepatitis. In particular, only ET reverted mitophagy and reduced mPTP opening [107]. Likewise, Rosa and colleagues detected an increase of autophagy (LC3-II/I ratio, p62) in mouse livers following a 4-week voluntary wheel running in both Western diet and normal diet groups, while Western diet suppressed BNIP3 levels by 30% compared to normal diet group. These authors proposed that increased autophagy may protect the liver from excessive lipid accumulation [108]. In addition, Tarpey found a remarkable increase of mitophagy in skeletal muscle biopsies from male runners after endurance training. However, they found no difference in mitophagy between fasting conditions and 4 h after high-fat diet intake, indicating that mitophagy may not be the dominant contributor to the exercise-induced protective metabolic flexibility against high fat diet intake [109]. One can argue that 4 h of high-fat diet intake is too short to impose any metabolic abnormality. To this end, these inconsistent findings are slightly biased, given the complexity of exercise and diet.

### 4.2. Mitophagy is Attenuated Due to Improved Mitochondrial Pool after Sustained Exercise Training

Mitophagy triggered by regular exercise precipitates the accumulation of healthy mitochondria as well as improved mitochondrial function. Therefore, mitophagy is believed to be maintained at an optimal (perhaps a low) level as a result of the long-term exercise training. Muscle biopsy obtained from human subjects showed increased LC3I, BNIP3, and Parkin levels 2 h following moderate cycling training. Interestingly, an increased capacity for mitophagy was also observed following an 8-week training [3]. Chen and coworkers further noted that sustained endurance training drastically attenuated exercise-induced mitophagy due to the overall improvement of mitochondrial quality [91]. Likewise, a study examining mitophagy between young and aged rat muscles revealed upregulated mitophagy in the aged group, while chronic contractile activity (CCA) limited mitophagy and improved mitochondrial stabilization [110]. In the same vein, another independent study also documented decreased mitophagy after a 5-day CCA. They further detected increased lysosome biogenesis regulator TFEB and LAMP1, indicating improved lysosomal degradation capacity [111]. Li and associates found that exhaustive exercise following exercise preconditioning displayed an unchanged LC3-II/LC3-I ratio. They further determined the levels of autophagy in different phases. Exhaustive exercise (EE) showed reduced LC3-II/LC3-I ratio, while exercise preconditioning (EP) transiently activated autophagy (especially at 2 h after EP) and attenuated EE-induced myocardial injury, which indicated preserved basal autophagy might underlie EP-offered benefit [112]. Besides endurance exercise, a study conducted by Estebanez and colleagues depicted that 8-week resistance exercise training prevented activation of mitophagy in peripheral blood mononuclear cells from otherwise healthy elderly individuals [113].

Ample studies have focused on the long-term effects of exercise on diseases as summarized nicely in recent reviews [1,3,100]. It is assumed that improved mitochondrial quality after exercise confers better cardiac performance and restrains pathological activation of mitophagy in response to acute stresses. Several attempts have been made to clarify whether exercise preconditioning imposes protective effects under acute cardiac stress. It has been shown that late exercise preconditioning protected the heart from exhaustive exercise-caused injuries through increasing Parkin-mediated mitophagy [114]. It was further suggested that exercise preconditioning augmented mitophagy via H_2_O_2_ oxidative stress-induced activation of PI3K [115]. Consistent with this view, it was found that earlier aerobic exercise complemented by a natural herb Rhodiola sacra protected the cardiac and skeletal muscles in exhaustive exercise through enhanced mitophagy [116]. Moreover, it has been demonstrated that exercise preconditioning may also exert protection against doxorubicin-induced cardiotoxicity [117]. Marques and team suggested that endurance exercise training before or during sub-chronic doxorubicin treatment prohibited doxorubicin-induced mitophagy, mPTP opening and apoptosis [118]. However, in contrast to finding from Marques, Lee argued that endurance exercise training prior to doxorubicin-treatment turned on protective mitophagy and suppressed NADPH oxidase 2 (NOX2) to protect against doxorubicin-induced cardiotoxicity [119].

Arrhythmia, especially fibrillation serves as a hallmark of cardiac injury and contributes to high cardiac mortality. Although a tight correlation between mitophagy and ischemic injury has been extensively described [120], whether mitophagy/autophagy participates in myocardial arrhythmia remains somewhat elusive. Lekli and associates thoroughly examined the occurrence of autophagy as an adaptive response to arrhythmogenesis, which might improve myocardial recovery through offsetting proteotoxic stress [121]. These authors suggested that intervention targeting autophagy should be taken with the precaution since excessive autophagy may be detrimental. A more recent study showed complex alternations of autophagy-associated proteins (decreased p62 and gradually reduced LC3BII/LC3BI) in ventricular fibrillation [122]. Thus, the association between arrhythmia and autophagy is unclear and further studies should be warranted. Isoproterenol is an extensively employed non-selective β-adrenergic agonist. It was suggested that at small dose of isoproterenol, autophagy may cope with the toxic arrhythmic effect of isoproterenol [123]. It is anticipated that non-invasive interventions such as exercise might be the countermeasure to arrhythmogenesis.

### 4.3. Compromised Mitophagy Response under Certain Pathological Conditions

It is possible that aging or certain metabolic diseases, such as obesity and diabetes, might compromise the regulation of mitophagy during exercise (shown in Figure 1) [58,124]. To examine whether mitophagy response induced by exercise vary in pathological states, a number of studies have been carried out. It has been shown that mitophagy flux stimulated by exercise was attenuated with age, resulting in mitochondrial deficiency during exercise in aging muscles [91]. Lipidated LC3II, the gold-standard indicator of autophagosome content, was upregulated 48 h following resistance exercise in untrained young but not older men [125]. The unfolded protein response (UPR) is another important adaptive reaction to exercise. Transcriptomic analysis revealed that the activation of UPR was attenuated in older healthy women and men compared to young adults following a single bout of exercise. Furthermore, the coordination between UPR and p53/p21 axis of autophagy was less evident in older [126]. In another independent study, an aging-induced significant decline of mitochondrial quality control proteins, such as Lon, could be partly rescued by exercise training [127]. Likewise, despite the increase of mitochondrial complex II, there was no noticeable change in BNIP3, MUL1, and LC3 II/I ratio in muscle biopsies of type 2 diabetic patients following a 3-month endurance training [128]. Nonetheless, contradictory findings are observed in human and rodent studies. A study tested mitophagy in mouse and human skeletal muscles. The results showed an aging-associated decline of PCG-1α and an increase of BNIP3 and LC3 II in mice, which was ameliorated by lifelong exercise training. However, markers of mitophagy and apoptosis were altered slightly during human aging, while lifelong exercise training upregulated BNIP3 [129]. It has been reported that a bout of unaccustomed resistance exercise for knee extensors transiently reduced the overall expression of mitochondrial proteins except for PCG-1α with no apparent change of mitophagy (VDAC, PINK1/Parkin) in both young and age candidates [130].

Sex difference exists in cardiovascular function [131,132]. Likewise, sex difference has also been noted in cardiac responses to exercise in individuals with cardiovascular diseases. Despite a similar exercise capacity, female heart failure patients with preserved ejection fraction (HFpEF) exhibited greater cardiac and extracardiac deficits, including worse biventricular systolic reserve, diastolic reserve, and peripheral O_2_ extraction [133]. A number of scenarios have been postulated for the sex difference in exercise-induced cardiovascular responses. For example, sex steroid hormones and their receptors exist in mitochondria from skeletal muscles, which may contribute to sex differences in cardiac performance in response to exercise. It was suggested that estrogen receptor binding attenuates the reduction in mitochondrial size and thus inhibits apoptosis. Other mechanisms in sex difference in exercise response may encompass activation of PI3K/AKT pathway and extracellular signal-regulated kinase 1/2 (ERK 1/2), which are also important regulators in exercise-induced mitophagy. Lack of estrogen and disruption of estrogen receptors might explain, in part, the reduced mitochondrial density and muscle mass in postmenopausal women [134]. Nonetheless, whether autophagy directly participates in these sex-related differences in exercise response remains unclear.

Moreover, studies have cast doubts on exercise-induced mitophagy. Unlike previous studies, Schwalm and colleagues provided evidence that mitophagy remained unchanged during and early (1 h) after acute high-density (70% VO_2peak_) endurance exercise in human skeletal muscles, whereas proteins and mRNA markers for mitochondrial fission and mitophagy (Drp1, Fis1, BNIP3) were more expressed in the fed state than the fasted state [135]. For some reason, a study including 11 participants examined gene expression of human muscles after exercise and argued that PINK1 and PARK2 mRNA were transiently decreased 3 h after 60-min cycling and returned to baseline 6 h later. These investigators also noticed that PCG-1α was elevated after exercise but was gradually decreased (albeit not below the baseline level) 6 h later [136]. In addition, a recent study also showed reduced mitochondrial mass and impaired respiratory function along with exercise-induced mitophagy induction in rat soleus muscles. However, the sample size was relatively small and the mitochondrial defect may also be attributed to lack of mitophagolysosome degradation [94]. Taken together, further research is needed to clarify the transient changes of mitophagy in health and diseases during and after exercise as well as how it impacts cellular health.

## 5. Conclusions

Given the ever-growing public concern on cardiometabolic diseases, there is an urgent need to hunt for effective preventive regimen, from pharmacological and non-pharmacological perspectives [137,138,139,140]. Given that physical inactivity is a well-known independent risk factor for all-cause mortality, regular physical exercise may offer profound health benefits in many aspects including cardiac performance, exercise tolerance, endothelial function, inflammatory response, insulin sensitivity, autonomic regulation, and blood pressure control along with glucose and lipid metabolism, adiposity, and psychosocial parameters [57,140]. Considering that exercise may impose both benefit and risk to human health, only modest or moderate exercise (less resistant type), is recommended to achieve a cardiovascular benefit. It is well perceived that regular moderate exercise may serve as an essential measure for the prevention and management of chronic diseases, including obesity, diabetes mellitus, atherosclerosis, and coronary artery disease [141,142]. Long-term exercise instigates physiological cardiac hypertrophy with preserved pump function. In this regard, a better understanding of the cellular and molecular mechanisms behind cardiac responses to exercise (physiology or pathological) should offer potential novel therapies against various cardiac anomalies. Given the critical role of mitochondria in the maintenance of cardiac homeostasis, mitochondrial quality control in particular mitophagy should be vital for cardiac health. In view of all that has been discussed in our review, we may propose that endurance exercise training protects cardiovascular system from acute stress possibly through maintaining homeostatic mitophagy. However, what we have learned about exercise-induced mitophagy is essentially based upon experimental studies and mainly skeletal muscles. There is a current paucity of well-controlled studies describing how exercise impacts cardiovascular function through regulation of mitophagy. To unveil the benefit versus risk for physical exercise on cardiovascular function, future studies should examine various types of exercise on autophagy and selective autophagy levels in an effort to provide insights into novel therapeutic avenues for the management of cardiovascular diseases. These findings will help us to evaluate the potential of mitophagy as a target for cardioprotection.

## Figures and Tables

**Figure 1 cells-08-01436-f001:**
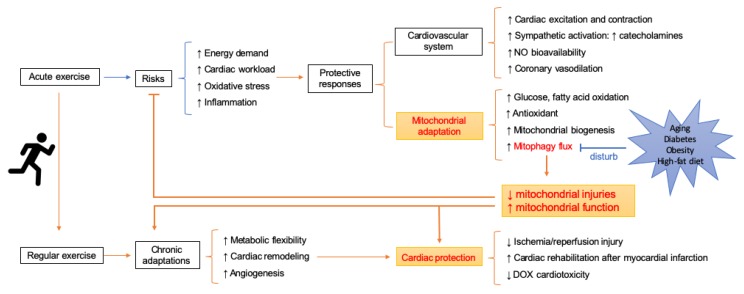
Schematic of mitochondrial adaptations in response to exercise and how it contributes to cardioprotection.

**Figure 2 cells-08-01436-f002:**
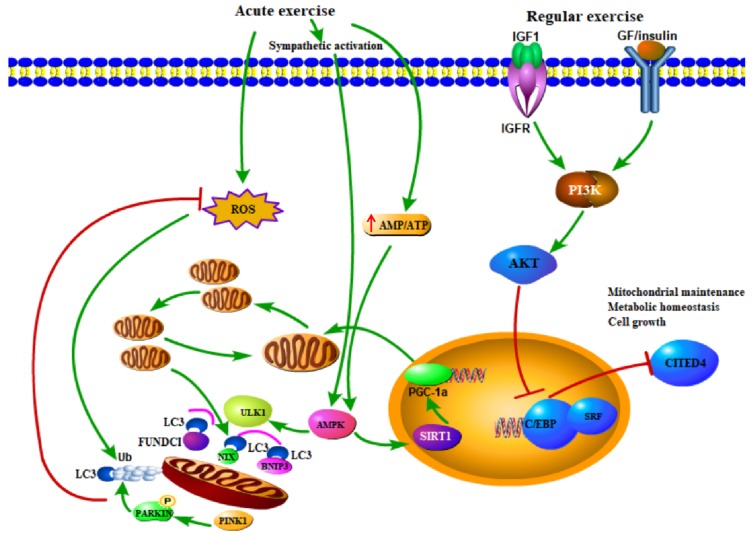
Mechanism and signaling pathways involved in mitochondrial adaptation in heart following exercise. Acute exercise augments mitophagy depending on the phosphorylation of AMPK (protein kinase AMP-activated catalytic subunit alpha 1) and ULK1 (unc-51 like autophagy activating kinase 1). AMPK could be activated by exercise-related increase of AMP/ATP ratio, sympathetic activation and other signaling. Mitophagy removes dysfunctional mitochondria and reduces reactive oxygen species (ROS). AMPK also promotes mitochondrial biogenesis through regulating PGC-1a. Regular exercise mainly activates the IGF1-PI3K-Akt pathway, which targets several transcription factors in nucleus and contributes to cell growth, cellular survival, metabolic homeostasis, and mitochondrial maintenance. Abbreviations: AMPK, AMP-activated kinase; Sirt1, Sirtuin 1; PGC-1a, peroxisome proliferator activated receptor gamma co-activator 1a; IGF-1, insulin-like growth factor-1; PI3K, phosphoinositide-3 kinase; Akt, serine/threonine-protein kinase; C/EBPβ, CCAAT/enhancer binding protein b; Cited4, cbp/p300- interacting transactivator with Glu/Asp-rich carboxy-terminal domain 4; SRF, serum response factor.

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
