# Peer review of "Physical Exercise and Selective Autophagy: Benefit and Risk on Cardiovascular Health"

_cells, 2019, doi:10.3390/cells8111436_

Round 1

Reviewer 1 Report

Comments the authors:

Ne N. Wu et al. summarized the connection between physical exercise and selective autophagy by careful consideration including benefits and risks on cardiovascular system. The topic is informative and a number of previous studies support these findings.

The following corrections should be added in the revised version of this review:

In section 2.4. The authors describe some additional knowledge about the crucial role of nitric oxide (NO) and nitric oxide synthase (eNOS) in the cardiovascular system. Please, include the function of NO and eNOS in the section 4 e.g., based on the previously published paper (doi: 10.1152/ajpheart.00561.2018. Please, also include (line 262), in the revision, that the normalized housekeeping proteins should be handled with caution under hypoxic conditions in the myocardial tissue, which could be in connection with autophagic processes (doi: 10.1155/2018/5786742.) Please, include, in the section 4.1., and discuss the importance of autophagy from another view as described by doi: 10.4149/BLL_2019_044. Could you mention, please, other exercise preconditioning-related publications in the section 4.2., e.g., line 348 (doi: 10.1536/ihj.18-310.). The authors also described, in section 4.3., that “It is possible that aging or certain metabolic diseases, …. might compromise the regulation of mitophagy during exercise”. Could the authors cite additional papers, which were published the compromised mitophagy responses under aging? Would be also useful to add a short section about the sex difference-related cardiovascular consequences of autophagy. It is also suggested to improve the revised version of this manuscript, if an additional section including the importance of arrhythmogenesis and the autophagy in the myocardium. Thus, the importance of the following three papers might be also discussed in a separated paragraph in the revised version of the review (doi: 10.3390/ijms20071628. PMID:30986903; doi: 10.2174/1381612825666190619145025. PMID:31258063; doi: 10.1111/jcmm.13053. PMID: 27997746).

Author Response

Ne N. Wu et al. summarized the connection between physical exercise and selective autophagy by careful consideration including benefits and risks on cardiovascular system. The topic is informative and a number of previous studies support these findings.

Response: Thank you for your insightful suggestions for our submission. We sincerely appreciate these valuable comments and have revised the manuscript accordingly. A point-by-point response is provided below. Changes to the manuscript are shown in red.

The following corrections should be added in the revised version of this review:

In section 2.4. The authors describe some additional knowledge about the crucial role of nitric oxide (NO) and nitric oxide synthase (eNOS) in the cardiovascular system. Please, include the function of NO and eNOS in the section 4 e.g., based on the previously published paper (doi: 10.1152/ajpheart.00561.2018. 

Response: This is a valid point. In the revision, we provided more information about NO and eNOS in the section 2.4 as follows “More recent evidence suggested that rhythmic handgrip exercise promoted increased eNOS phosphorylation, NO generation, and O2- production, along with improved autophagy markers including Beclin1, microtubule-associated proteins 1A/1B light chain 3B (LC3B), autophagy-related gene 3 (Atg3), and lysosomal-associated membrane protein 2A (LAMP2) as well as decreased levels of p62 in endothelial cells from human radial artery [51]. These findings denote a close tie between eNOS/NO signal cascade and autophagy in exercise-induced regulation on endothelial function”. We have also expanded section 4 on this topic as follows “It’s noteworthy that shear stress has emerged as a modulator of autophagy during exercise. It was reported that 1 hour of rhythmic handgrip exercise initiated autophagy, NO generation and O2·- production in humans due to elevation of shear stress [51]”.

Please, also include (line 262), in the revision, that the normalized housekeeping proteins should be handled with caution under hypoxic conditions in the myocardial tissue, which could be in connection with autophagic processes (doi: 10.1155/2018/5786742.)

Response: Thanks much for alerting us this and it has been inserted (line number 267-270 in the revision).

 Please, include, in the section 4.1., and discuss the importance of autophagy from another view as described by doi: 10.4149/BLL_2019_044.

Response: Thanks for this input. This helpful paper has been added (line 311-313 in the revision).

Could you mention, please, other exercise preconditioning-related publications in the section 4.2., e.g., line 348 (doi: 10.1536/ihj.18-310.). 

Response: This is a valid point and we included more information about exercise preconditioning (the suggested reference being added) (line 355-360 in revision).

The authors also described, in section 4.3., that “It is possible that aging or certain metabolic diseases, …. might compromise the regulation of mitophagy during exercise”. Could the authors cite additional papers, which were published the compromised mitophagy responses under aging?

Response: Thank you for the suggestion. We have added more references to support this idea (line 398-405).

Would be also useful to add a short section about the sex difference-related cardiovascular consequences of autophagy.

Response: We sincerely appreciate the valuable comment. We have checked the literatures carefully and added more references on sex difference (line 416-427).

It is also suggested to improve the revised version of this manuscript, if an additional section including the importance of arrhythmogenesis and the autophagy in the myocardium. Thus, the importance of the following three papers might be also discussed in a separated paragraph in the revised version of the review (doi: 10.3390/ijms20071628. PMID:30986903; doi: 10.2174/1381612825666190619145025. PMID:31258063; doi: 10.1111/jcmm.13053. PMID: 27997746). 

Response: This issue (arrhythmia, exercise and autophagy) has been emphasized in the revision (line 379-391).

Reviewer 2 Report

It is an enjoyable article on the pathophysiological role of mitophagy as molecular mechanism of physical exercise adaption. It underlines some interesting and yet debated mechanisms and, above all, it does stimulates clinicians and researchers to evaluate this tangled and fascinating topic.

Author Response

Response: Thank you for your kind comments on our manuscript. These comments are all valuable and helpful for improving our article.

Reviewer 3 Report

The review highlights an interesting topic. Selective autophagy in the context of physical exercise has not been thoroughly described yet. The authors provide a review, which is well written and structured. It constitutes a significant expansion of the current knowledge. The manuscript should be edited regarding English language. Furhter, the authors might provide an outlook of future work on this topic.

Author Response

Response: Thank you for your insightful suggestion. We have tried our best to improve the revision and made substantial changes accordingly. These changes should greatly improve the overall depth of the manuscript without affecting the framework of the original submission. All changes made are marked in red color in the revised manuscript. We sincerely appreciate for the valuable comments earnestly, and hope that the revised sections will be satisfactory.

Round 2

Reviewer 1 Report

No additonal comments. Accept as is.